# Soil Fertilization with Palm Oil Mill Effluent Has a Short-Term Effect on the Bacterial Diversity of an Amazonian Agricultural Land Area

**DOI:** 10.3390/microorganisms12030507

**Published:** 2024-03-01

**Authors:** Johnes Pinto Sanches, Sávio Souza Costa, Diego Assis das Graças, Artur Silva, Guilherme Costa Baião, Rennan G. Moreira, Marcelo Murad Magalhães, Roberto Lisboa Cunha, Rafael Azevedo Baraúna

**Affiliations:** 1Biological Engineering Laboratory, Innovation Space, Guamá Science and Technology Park, Belém 66075-750, Brazil; johnes.sanches@icb.ufpa.br (J.P.S.); savscosta@gmail.com (S.S.C.); diego.a87@gmail.com (D.A.d.G.); arturluizdasilva@gmail.com (A.S.); 2Laboratory of Molecular and Computational Biology of Fungi, Department of Microbiology, Institute of Biological Sciences, Federal University of Minas Gerais, Belo Horizonte 30120-140, Brazil; gcbaiao@gmail.com; 3Multiuser Laboratories Center, Center of Genomics, Institute of Biological Sciences, Federal University of Minas Gerais, Belo Horizonte 31270-901, Brazil; rennangm@gmail.com; 4Brazilian Agricultural Research Corporation (EMBRAPA), Eastern Amazon, Belém 66095-100, Brazil; marcelo.magalhaes@embrapa.br (M.M.M.); roberto.cunha@embrapa.br (R.L.C.)

**Keywords:** 16S rRNA, bioinformatics, POME, microbiome, palm oil, metabarcoding

## Abstract

Palm oil derived from the fruits of *Elaeis guineensis* Jacq. has global economic importance and is largely produced in tropical regions. The palm oil production process leads to a highly polluting waste called palm oil mill effluent (POME). A strategy commonly used by producers to overcome environmental issues and to improve soil fertility is the reuse of POME as a fertilizer due to the chemical and biological characteristics of the effluent. In this research, three groups were analyzed: soil without POME application (control group) and soil samples after 4 and 9 days of POME application. An environmental DNA metabarcoding approach was used. eDNA was extracted, and the V4 region of the 16S rRNA gene was amplified and sequenced in the Illumina MiSeq platform. The abundance of Proteobacteria (48.1%) and Firmicutes (9.0%) was higher in fertilized soil, while Bacteroidetes (20.3%) and Verrucomicrobia (7.8%) were more abundant in control soil. Additionally, the effluent seemed to modify soil characteristics favoring taxa responsible for the mineralization of organic compounds and nitrogen fixation such as species of *Gammaproteobacteria* class. Our study highlights the influence of POME on soil biological components and contributes to the sustainable production of palm oil in the Amazon.

## 1. Introduction

*Elaeis guineensis* Jacq., the oil palm, is a perennial crop native to equatorial Africa that was introduced in other tropical regions of the world such as the Brazilian Amazon. The oil extracted from the palm fruits is used in several products including food, cosmetics, and biodiesel. Oil palm has been used as a semi-wild food by traditional communities for more than 7000 years [1]. In South America, the plant is also used in traditional medicine by indigenous people [2]. The industrial farming of oil palm remains predominantly concentrated in Southeast Asia. However, other tropical regions in the world have significantly expanded their production, including West Africa and South America [1].

*Elaeis guineensis* Jacq. offers an extracting yield significantly higher than other palm trees and presented a worldwide production of 77.2 million tons in the crop year 2022/2023 [3,4]. Brazil represents 15% of this worldwide production, with 11.4 million tons produced in the same period [4]. The state of Pará has the largest agricultural land area, producing 98% of the exported oil [5]. The Amazon region offers ideal ecological conditions for oil palm farming. Over the last five decades, this has become one of the most important crops for the region’s economy [6]. Two types of oil can be obtained from the fruit: palm oil, extracted from the mesocarp, represents 89% of the total, and palm kernel oil, extracted from the seed or almond, represents the remaining 11% [7].

The economic and social importance of this commodity is unquestionable; however, palm oil extraction leads to the massive production of a liquid waste called palm oil mill effluent (POME). About 5 to 7.5 tons of water are necessary to produce 1 ton of palm oil, and approximately 50% of the water is converted into POME [8].

The POME is a colloidal suspension, characterized by its viscous and brownish liquid containing approximately 95–96% water, 0.6–0.7% oils, and 2–4% suspended solids originated from the decomposition of fruit residues; this substance stands out for its non-toxic nature [9]. POME has a high concentration of organic matter, and its untreated release into water bodies can cause pollution, deplete oxygen levels, and impact aquatic ecosystems. Sharuddin and colleagues [10] demonstrated that POME discharge into riverbeds not only impacts the water’s physicochemical characteristics but also the microbial diversity of the receiving water body. In fact, POME presents high rates of biochemical oxygen demand (BOD), varying from 18,000 to 48,000 mg L^−1^ [11]. The maximum BOD value allowed by the Brazilian legislation for the safe environmental disposal of effluents is 120 mg L^−1^. If this threshold is exceeded, it is necessary to use an effluent treatment system with a minimum BOD removal rate of 60%. Indeed, this environmental issue represents a major challenge that raises uncertainty about the long-term ecological viability of the global palm oil industry. Thus, it is crucial to adopt sustainable practices to face environmental challenges, preserve ecosystems, and promote economic responsibility in the industry. Therefore, the high production of POME represents a major challenge for the palm oil production chain.

A strategy widely used by producers to overcome these environmental issues consists of reusing POME as a soil fertilizer, due to its rich nutrient content, particularly nitrogen and phosphorus, and microbiological elements contained in the effluent. POME harbors diverse bacterial communities, being a source of microorganisms with potential biotechnological application [12,13,14]. The microbial diversity of POME varies depending on whether the palm oil extraction process is handmade or industrial [12]. The effects of discharging this effluent into soils have been studied in different biomes and under different conditions. Most research uses cultivation-dependent techniques, which limits the ecological analysis of the microbial community [15].

Environmental DNA (eDNA) metabarcoding studies that use 16S rRNA gene sequencing allow accurate ecological inferences in order to explore the taxonomic and functional diversity of microbial communities in different environmental samples [16,17]. The 16S rRNA gene has a size of approximately 1500 bp and contains hypervariable regions that are used for the taxonomic affiliation of Operational Taxonomic Units (OTUs) or Amplicon Sequence Variants (ASVs) [18]. Some protocols amplify one or two hypervariable regions, producing amplicons of around 200–400 bp. Recently, long-read sequencing techniques such as nanopore-based platforms have allowed the analysis of the complete gene, which considerably improved the accuracy of taxonomic classification [19,20].

Recently, several studies have been conducted to make effluent treatment systems more economical and effective, overcoming the limitations of conventional ponds. The widespread use of these ponds is justified by low initial investment, minimal technical requirements, and reduced operational costs [21]. Management techniques are being analyzed to develop affordable and effective methods, with the goal of effectively reducing chemical oxygen demand (COD) and BOD [22]. Among these techniques, low-cost unconventional methods stand out, which include microbial degradation processes, physicochemical coagulation processes, membrane filtration, and thermochemical procedures [23]. These innovative approaches aim to optimize the treatment of POME, providing greater efficiency and sustainability.

Regarding microbial processes, there is a lack of information on the microbial diversity of POME produced in the Amazon region and few studies evaluating the effects caused by soil fertilization with POME. Therefore, to ensure the sustainable management of Amazonian agricultural land, it is of great importance to evaluate how this effluent interacts with the biological components of the soil. Thus, our work aimed to evaluate the impact of POME fertilization on soil bacterial diversity in an Amazonian agricultural land area.

## 2. Materials and Methods

### 2.1. Sampling and DNA Extraction

The palm oil tree is a perennial crop plant. The sampled areas of *E. guineensis* Jacq. are 11 years old, located in the municipality of Mojú, Pará, Brazil (1°58′43.2″ S and 48°36′52.8″ W), and belong to a private company. The region has an average annual temperature and precipitation of 25 °C to 27 °C and 2000 mm to 3000 mm, respectively. Additionally, the region has an irregular rainfall pattern, with a short period of drought [24]. The soil of the study area is classified as sandy texture yellow latosol. Only the soil microbiome analyses were approved by the company; therefore, this manuscript does not present soil physicochemical data.

Sampling was performed in two distinct areas during September 2022, corresponding to the Amazon dry season. The first area was composed of soil not fertilized with POME (SP) (control group). Six replicates of SP were sampled at a depth of 20 cm, 1.5 m away from the palm tree (Appendix A). Replicates were sampled near different palm trees planted in an equilateral triangle crop system with an average distance of 9 m between trees. The second area contained soil fertilized with POME (test group). POME was sterilized before fertilization. POME was applied to the soil with a 10,000 L agricultural tank coupled to a tractor. Ten tons of POME per hectare was used, and the application was performed in the middle of the tree lines. Sampling was performed 4 (P4) and 9 (P9) days after POME application. The sampling design was the same as that used for the control area. Six replicates were sampled for each group (P4 and P9). Thus, a total of 18 soil samples were analyzed. Soil samples were stored in sterile polypropylene vials at 4 °C until DNA extraction, which was performed on the same day. Environmental DNA was extracted using the FastDNA SPIN kit for Soil (MP Biomedicals, Solon, CA, USA) according to the manufacturer’s instructions. DNA quantification was performed in Qubit 2.0 (Thermo Fisher Scientific, Waltham, MA, USA), and DNA integrity was evaluated using 1% agarose gel. Extracted DNA was stored at −20 °C until sequencing.

### 2.2. 16S rRNA Gene Sequencing

The 16S rRNA gene was amplified using the primers 515 F (5′-TCG TCG GCA GCG TCA GAT GTG TAT AAG AGA CAG GTG CCA GCM GCC GCG GTAA-3′) and 806 R (5′-GTC TCG TGG GCT CGG AGA TGT GTA TAA GAG ACA GGG ACT ACH VGG GTW TCT AAT-3′) which amplify the hypervariable region V4. Reactions were performed for a final volume of 20 µL containing 12 ng of DNA, 4 µM of each primer, 0.2 mM of dNTPs, 2.5 mM of MgCl_2_, and 0.5 U of DNA polymerase Phusion Hot Start II. The PCR steps were initial denaturation at 98 °C for 30 s, followed by 25 cycles of 98 °C for 30 s, 55 °C for 60 s, and 72 °C for 30 s, with a final extension step of 72 °C for 5 min. Amplicons were indexed, quantified, and sequenced on the Illumina MiSeq platform using the Nano kit v. 2 (Illumina, San Diego, CA, USA) (300 cycles), paired reads (2 × 250 bp), according to the manufacturer’s instructions.

### 2.3. Bioinformatics

Raw data were analyzed using Divisive Amplicon Denoising Algorithm (DADA2) v. 316 [25]. Low-quality bases (Q < 20) were trimmed using parameters truncLen = c (248, 248), maxN = 0, max EE = c (3, 4), truncQ = 2, rm.phix = TRUE, compress = TRUE, multithread = FALSE. Reads were subsequently merged, and chimera sequences were removed. Taxonomic affiliation was performed through the Amplicon Sequence Variants (ASV) method using the SILVA database v. 128 [26].

Alpha- and beta-diversity analyses were performed in R v. 4.3.1 using the phyloseq v. 1.22.3 and vegan v. 2.6-4 packages [27,28]. Alpha-diversity indexes such as Chao, Shannon, and Simpson were calculated and compared using PERMANOVA and the Kruskal–Wallis test. Bray–Curtis distance and unweighted and weighted Unifrac were used for beta-diversity analysis after rare taxa removal (<1%). Other plots such as heat maps and rarefaction curves were obtained using standard parameters of the packages.

## 3. Results

### 3.1. Alpha and Beta Diversity

A total of 450,008 raw 16S rRNA gene sequences were produced and analyzed. Sequencing metrics per sample are presented in Appendix A. The samples were divided into three groups: soil without POME application (SP), soil after 4 days of POME application (P4), and soil after 9 days of POME application (P9). One of the SP replicates showed low DNA extraction yield and was therefore excluded from sequencing. After quality assessment and trimming, the number of sequences per sample varied from 17,235 to 37,435. The reads presented an average size of 251 bp. Samples 14P9 and 6P4 presented higher and lower numbers of predicted ASVs, respectively (Appendix A).

Our culture-independent analysis showed a significant shift in bacterial taxonomic composition comparing unfertilized and POME-fertilized soils. No significant difference in bacterial diversity was observed comparing samples P4 and P9. Beta diversity is presented in a Principal Component Analysis (PCoA) plot based on Bray–Curtis distance values (Figure 1A), which also demonstrates clustered P4 and P9 samples and more distant SP samples.

The Chao1, Shannon, and Simpson diversity indexes were higher in POME-fertilized soils (P4 and P9) compared to control samples (P4: *p*-value = 0.002, F = 7.36, PERMANOVA; P9: *p*-value = 0.007, F = 5.26, PERMANOVA) (Figure 1B–D). It was possible to observe that the bacterial composition of the SP samples was significantly different from that of the P4 and P9 samples.

### 3.2. Taxonomic Composition of Microbial Communities

Our sequencing effort was sufficient to describe bacterial diversity (Figure 2A) once rarefaction curves reached a plateau. It was possible to observe that P9 samples showed higher species richness (Figure 2A). In total, 41 phyla, 197 families, and 3287 genera were described in the dataset. ASVs without taxonomic affiliations were defined as unclassified microorganisms and represented 2.3% of the dataset. The most abundant phyla were Proteobacteria, Bacteroidetes, Firmicutes, and Verrucomicrobia (Figure 2B). Bacteroidetes and Verrucomicrobia were abundant in samples without POME application, while Proteobacteria and Firmicutes were abundant in the P4 and P9 samples (Figure 2B). Thus, a shift in bacterial composition was observed after POME application. Regarding bacterial classes, *Gammaproteobacteria*, *Sphingobacteriia*, *Clostridia*, *Spartobacteria*, *Flavobacteriia*, *Bacilli*, and *Alphaproteobacteria* were the most abundant. *Gammaproteobacteria* prevailed in the P4 and P9 samples, while *Sphingobacteriia* was abundant in the SP samples (Figure 2C). At the family level, *Pseudomonadaceae*, *Clostridiaceae 1*, DA 101 soil group, *Chitinophagaceae*, and *Moraxellaceae* were the most abundant (Figure 2D), with *Pseudomonadales*, *Clostridiaceae*, and *Moraxellaceae* being most prevalent in soil fertilized with POME, while *Chitinophagaceae* and DA 101 soil group were the most abundant in samples without POME application.

The heatmap (Figure 3A) shows the significant difference in phyla abundance among fertilized and unfertilized soils. The high abundance of Proteobacteria in POME-fertilized soil is mainly due to an increase in abundance of the *Azotobacter*, *Acinetobacter*, and *Geobacter* genera (Figure 3B). Similarly, the increase in Firmicute abundance is mainly due to an increase in the abundance of the *Clostridia* class (Figure 3C). No significant change was observed in the Verrucomicrobia and Bacteroidete richness despite both showing a decrease in abundance compared to control soil (Figure 3A and Appendix A). *Spartobacteria* and OPB 35 soil group were the most abundant Verrucomicrobia classes, while *Sphingobacteriia*, *Flavobacteriia, Bacteroidia*, and *Cytophagia* were the most abundant Bacteroidete classes, with *Bacteroidia* most prevalent in soil that received POME application (Appendix A).

## 4. Discussion

In this study, a culture-independent method based on the massive sequencing of the 16S rRNA gene was used to evaluate the impact of soil fertilization with POME on the bacterial diversity of *E. guineensis* farming areas located in the Amazon. This approach is considered the “gold standard” for evaluating the structure and diversity of environmental microbial communities, enabling the accurate identification of microbial diversity variations [29]. Soriano-Lerma and colleagues demonstrated that the choice of the hypervariable region for microbial diversity studies does not produce bias in taxonomic analysis at the genus level [29]. The microbial diversity of Amazonian soil has been analyzed from several perspectives, with a special focus on studies of deforestation that result in the conversion of rainforest-to-pasture or -agriculture. Regarding the oil palm industry, as far as we know, few microbial ecology studies have been performed in the region. In other regions of the world, eDNA metabarcoding studies were used to evaluate the dynamics of microbial populations during the biodegradation of POME in a full-scale treatment system [13] and to identify potential bioindicators of POME discharge in receiving water bodies [10], among other aims.

In Brazil, the disposal of POME in water bodies must follow CONAMA Resolution N° 430/2011. Therefore, the effluent used in this study presented a pH between 5 and 9, temperature < 40 °C, an absence of floating materials, and the minimum removal of 60% BOD for 5 days at 20 °C, among other parameters. The analyses were performed by the company responsible for the farming area.

Our data revealed that the application of POME to the soil for fertilization significantly changed the composition of bacterial communities. Soil fertilized with POME showed greater bacterial taxonomic diversity, a positive indicator of soil health. The low diversity observed in the soils of the control group is probably associated with the large-scale monoculture of perennial crops, a practice that leads to soil degradation [30]. Thus, the use of POME as a fertilizer seems to be an alternative to enrich the microbial community of these soils. No significant changes in microbial diversity indexes were observed between the 4th and 9th days after the application of POME. Thus, additional analyses are necessary to assess mid- to long-term modifications.

The most abundant bacterial phyla found in this study (Proteobacteria, Firmicutes, Bacteroidetes, and Verrucomicrobia) were the same as those described by Fadiji and colleagues [31] in soil with organic fertilization. Goh and colleagues [32] also described the Proteobacteria and Firmicutes phyla as the most abundant in Malaysian soil. However, several other phyla with considerable abundance were found, such as Acidobacteria and Chloroflexi [32]. Interestingly, the most abundant phyla in samples P4 and P9 were the same as those described in POME samples from industrial oil palm production systems in the Republic of Côte d’Ivoire [12]. The effluent used as fertilizer is sterilized before application to the soil. Therefore, it is unlikely that POME acts as a vehicle for the inoculation of exogenous bacteria. Instead, we believe that the effluent markedly modifies soil characteristics, favoring the growth of taxa adapted to these conditions such as those described in P4 and P9.

The *Gammaproteobacteria* class belongs to the Proteobacteria phylum and is directly related to organic matter degradation and soil nutrient cycling [33]. About 10.5% of the ASVs in samples P4 and P9 were affiliated to this class. Thus, *Gammaproteobacteria* are probably the main taxon involved in the mineralization of POME organic matter. The nitrogen-fixing genus *Azotobacter* was the most abundant gammaproteobacterial in POME-fertilized soil. The importance of this taxon and other members of the *Pseudomonadales* family in soil fertility has already been described in the scientific literature [34]. Other genera found in P4 and P9 samples that deserved attention were *Pseudomonas*, *Acinetobacter*, and *Perlucidibaca*. *Pseudomonas* are commonly found in the rhizosphere, helping plant growth through several mechanisms [35]. Some species colonize the plant root, preventing the action of phytopathogenic fungi and bacteria [36]. For example, semi-commercial formulations containing *Pseudomonas chlororaphis* PCL 1606 have demonstrated the ability to control *Rosellinia necatrix* infestation in avocado soil [37]. *Acinetobacter* species have been applied to remove herbicides and pesticides from the soil. *A. calcoaceticus* was able to remove Carbendazim, while *A. lwoffii* demonstrated potential for removing Atrazine [38,39]. Additionally, some phosphate-solubilizing *Acinetobacter* species, such as *A. pittii* gp-1, when inoculated into phosphorus-limited soil, promote soybean growth by increasing the absolute abundance of genes involved in phosphorus cycling and improving the availability of indole acetic acid [40]. Finally, a low number of ASVs were affiliated to *Perlucidibaca*. This genus was described by Teng and colleagues [41] as responsible for the degradation of pyrene in oil land soils. The ability to degrade complex hydrocarbons is certainly a powerful function in POME-fertilized soil.

Firmicutes was one of the most abundant phyla on POME-supplemented samples after 4 days, with a slight decrease after 9 days. It was mostly represented by Clostridia and Bacilli classes (90%), which are (strictly or facultative) anaerobes found in several environments, such as the human gut, marine sediments, and soil. These classes are frequently reported as able to hydrolyze complex organic compounds, breaking down large molecules into simpler forms [42]. In fact, our data show an increase in the Clostridia class during the first 4 days (P4), followed by a slight decrease after 9 days (P9). The Clostridia and Bacilli classes found in this study have been reported as beneficial for plant growth due their metabolic capabilities [43]. The dominance of Firmicutes was consistent with previous reports at different stages of POME treatment, encompassing anaerobic, facultative anaerobic, and aerobic processes [13].

Due to high concentrations of organic matter, suspended solids, and nutrients in POME composition, microbes degrade these compounds, depleting oxygen levels and creating an anaerobic condition [44], which is used as POME waste treatment to produce methane. Methanogenic archaea were not evaluated in this study, but it is known that the Clostridia, Bacilli, and Anaerolineaceae (Chloroflexi) found in this study can synergically act along methanogens in POME samples. It was reported that after 12 weeks of degradation, the co-compost of empty fruit bunch (EFB) and palm oil mill effluent (POME) showed an abundance of 18% of Anaerolineacea (25% Chloroflexi), and that anaerobes and fermenting bacteria dominated the microbiome [44]. Our data show less than 2% of Chloroflexi assigned sequences (many Anaerolineacea), which means that this group has a small role in the first days but could increase after a longer period. After the introduction of POME, bacteria that tolerate anaerobic conditions increased their abundance, which is expected when high organic matter content is added to an environment, but after 9 days, the community structure slightly decreased in microbial diversity (not significantly), which also requires long-term analysis.

In this study, we found a dramatic decrease in Verrucomicrobia abundance after 4 days of POME introduction, from 15% to 1.4% (mean). On the other hand, we have shown that Verrucomicrobia is quite resilient, increasing abundance (mean 9%) after 9 days. In fact, Verrucomicrobria is abundant and ubiquitous in soil samples, and in our study was represented by Spartobacteria (DA 101 group) and the OPB 35 soil group. DA 101 group abundance varies according to soil type/location and has aerobic heterotroph representants, which have relatively small genomes and are able to use few carbon sources, such as glucose, pyruvate, and chitobiose [45]. The OPB 35 group remains uncultured, and little is known about this clade, but it is abundant in wetland soils and sediments [46].

All taxa described above appear to have a beneficial effect on the soil, which emphasizes the role of POME in improving the soil fertility of monoculture crops. In contrast, 17.9% of the ASVs found in control samples were affiliated to the *Sphingobacteriia* class, which plays a major role in complex biopolymer degradation such as (hemi)cellulose [47]. The abundance of this taxon emphasizes its importance in the degradation of complex organic matter in the soil of monoculture crops.

Previous research has demonstrated the close relationship between soil quality and increased crop productivity. By incorporating beneficial microorganisms into the soil, this positive effect can be amplified [20]. Our results demonstrate that the reuse of POME as an agricultural fertilizer promotes a rapid increase in soil bacterial diversity, favoring the emergence of beneficial taxa such as *Azotobacter*, *Sphingobacteria*, *Bacilli*, *Clostridium*, *Pseudomonas*, *Acinetobacter*, *Azorhizophilus*, and *Perlucidibaca*. After four days, POME degradation consumes oxygen, and anaerobes have their growth favored, which remains after 9 days, but is reduced in abundance. However, additional studies are necessary to evaluate the physicochemical impacts as well as the ability of POME components to reach deeper soil layers. Our data showed that the rare microbiome (less than 1% abundant phyla) represented less than 5% in total, but many changes were detected. A more focused analysis on these rare taxa could enhance our knowledge about the introduction of POME on soil and its use as a fertilizer.

Many studies are still necessary before POME can be used as a fertilizer, and they should consider different soil types, crops, and the optimal amount. Other organisms such as soil methanogenic archaea and mycorrhizal fungi play crucial roles, not only in degrading organic matter from POME, but also in promoting plant growth and overall ecosystem health. This type of study aligns with sustainable agricultural practices by recycling waste materials to enhance soil fertility and avoiding water pollution. It represents a potential circular economy approach within the palm oil industry.

## 5. Conclusions

Our study analyzed the short-term impact of POME on the soil bacterial diversity in an Amazonian palm oil production area. POME application significantly changed the taxonomic composition. Proteobacteria and Firmicutes predominated in fertilized soils, while *Bacteroidetes* and *Verrucomicrobia* were more abundant in soils without the application of the effluent. Genera such as *Azotobacter*, *Acinetobacter*, and *Geobacter* were related to the increase in Proteobacteria abundance, while the *Clostridia* class was related to the increase in Firmicutes’ abundance. These results indicate that POME fertilization can positively modulate the soil bacterial community, contributing to the emergence of beneficial taxa for the sustainable production of palm oil in the Amazon. Further analyses are needed to determine the long-term stability of these changes.

## Figures and Tables

**Figure 1 microorganisms-12-00507-f001:**
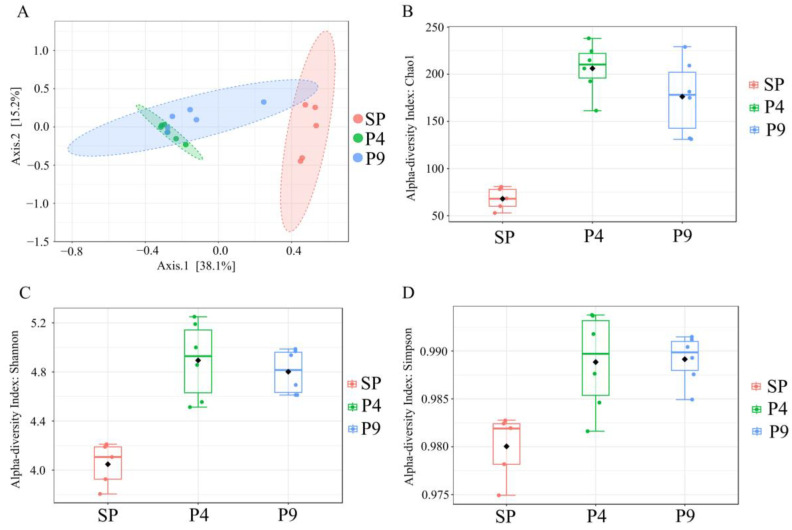
(**A**) PCoA plot showing beta-diversity analyses. (**B**) Chao1, (**C**) Simpson, and (**D**) Shannon diversity indexes for the three sample groups analyzed. SP = Samples without POME (*Palm Oil Mill Effluent)*; P4 = soil samples after 4 days of POME application; P9 = soil samples after 9 days of POME application.

**Figure 2 microorganisms-12-00507-f002:**
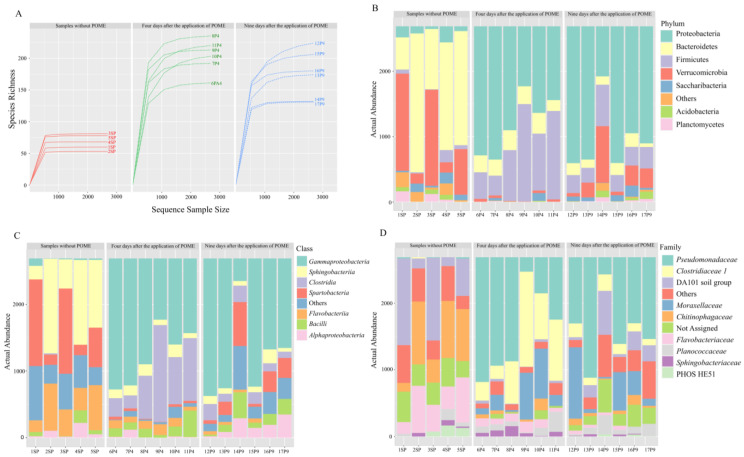
(**A**) Rarefaction curve and histogram showing the relative abundance of 16S rRNA gene amplicon reads assigned to level of (**B**) phylum, (**C**) class, and (**D**) family. Unclassified ASVs were shown as “not assigned”. The 10 most representative taxa were presented in the legends.

**Figure 3 microorganisms-12-00507-f003:**
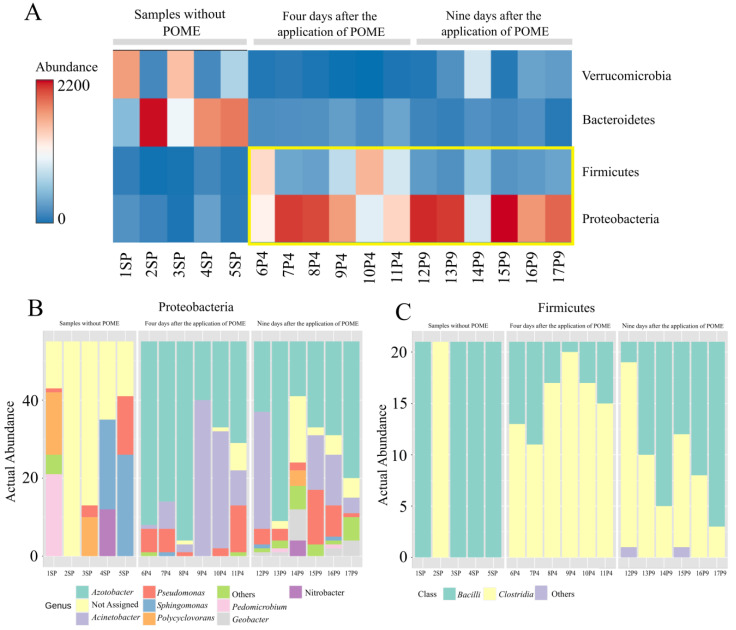
(**A**) Heatmap comparing the relative abundance of the Proteobacteria, Firmicutes, Bacteroidetes, and Verrucomicrobia phyla. The yellow square highlights the phyla with greater abundance in the area where POME was applied. Species richness assigned at the level of (**B**) genus or (**C**) class for the Proteobacteria and Firmicutes phyla, respectively.

## Data Availability

Data are available in the Sequence Read Archive (SRA) database under the accession number PRJNA1043058.

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
