# Peer review of "Soil Fertilization with Palm Oil Mill Effluent Has a Short-Term Effect on the Bacterial Diversity of an Amazonian Agricultural Land Area"

_microorganisms, 2024, doi:10.3390/microorganisms12030507_

Round 1

Reviewer 1 Report

Comments and Suggestions for Authors

This study tested the effects of Palim Oil  Mill Effluent (POME) application on soil microbial abundance and diversity through an environmental DNA metabarcoding approach in a farmland soil. The author found the POME altered the soil microbial abundance, such as the abundances of Proteobateria and Firmicutes were enhanced, while  the abundances of Bacteroidetes and Verrucomicrobia were decreased. However, there were some defects, which need to be improved. Why did the POME affect soil microbial abundance? No related data supported this conclusion. Please supply the information. Overall, the topic of this MS is interesting, yet this version of the MS should be improved greatly.

1. The soil taxa should be clarified in materials and methods section.

2. The microbial abundance from the POME should be provided.

3. Soil physicochemical properties in three treatments and the POME’s characteristics should be provided, especially soil available nutrients. They are vital to explain the alteration of soil microbial abundance and diversity.

4. 4 and 9 days of POME application were short time, thus this study only focused on the short-term effect. This information thus may be highlighted in the title.

Author Response

Dear Reviewer,

Kind regards,

Reviewer 2 Report

Comments and Suggestions for Authors

Dear Authors!

The manuscript is devoted to the actual topic of recycling industrial waste to increase the biological activity of the soil and promote sustainable agricultural practices. The manuscript may be of interest to readers, but in order to improve its quality, I propose to make the following changes to it.

1. Line 162. The table number was incorrectly specified (S1 instead of S2).

2. How much and how was ROME added to the soil? Has ROME been sterilized?

3. Why is there no soil characteristic (organic matter, microelements, pH, etc.)?

3. There is no reference to Table S1 in the text. What do ng/uL, % PF, Index 1 I7, Index 2 I5 mean?

4. Why was sampling carried out on days 4 and 9 after POME treatment? Why was the very short sampling interval chosen?

5. It is necessary to write the sampling methodology more clearly. For example, it is not clear why 5 samples SP and 6 samples each from P4 and P9 were analyzed. Figure S1 does not provide a clear picture of the Sampling design.

6. Fig. 3A. There was an error in the designation of samples taken on day 9 (12P6, 13P6, etc.).

7. How, according to the authors, will the composition and abundance of the bacterial community of the soil change after its treatment with POME after a longer period of time, for example, on days 14, 21, 30?

8. The manuscript is missing the Conclusion section.

Comments on the Quality of English Language

Minor editing of English language required

Author Response

Dear Reviewer,

Kind regards,
